# MRI2PET: Realistic PET Image Synthesis from MRI for Automated Inference of Brain Atrophy and Alzheimer's

## Abstract

Positron Emission Tomography (PET) is a crucial tool in medical imaging diagnostics but remains costly and less accessible than alternatives like X-Ray and MRI. To address this, we propose `MRI2PET`, a 3D diffusion-based model that generates AV45-PET scans from T1-weighted MRI images. `MRI2PET` incorporates style-transferred pre-training and a Laplacian pyramid loss to leverage unpaired MRI data and structural correspondences between modalities while simultaneously emphasizing the crucial details. Using the ADNI dataset, we demonstrate that `MRI2PET` produces realistic PET images and improves downstream clinical classification. Notably, augmenting the original PET-only training data with `MRI2PET`-synthesized scans increases AUROC from $0.688 \pm 0.014$ to $0.780 \pm 0.005$ when classifying into one of cognitively normal, mild cognitive impairment, and Alzheimer's Disease groups. These results highlight `MRI2PET`'s ability to generate high-quality, clinically informative PET scans from widely available MRI, offering an accessible, cost-effective approach to enhance machine learning performance and expand diagnostic imaging workflows.

## 1 Introduction

Positron Emission Tomography (PET) scans represent a cornerstone of advanced medical imaging diagnostics. These molecular imaging studies provide crucial insights into metabolic processes (Kapoor & Kasi, 2020), enabling precise detection of cancer (Jerusalem et al., 2003; Pimiento et al., 2016; Vansteenkiste & Stroobants, 2006), detailed evaluation of brain function (Myers et al., 1996), and comprehensive assessment of cardiac health (Hartiala & Knuuti, 1995; Peretto et al., 2022). Specifically, their ability to provide nuanced clinical details, such as precise tumor metabolic activity, makes them invaluable for personalized treatment planning (Schrevens et al., 2004; Vansteenkiste, 2003). However, the widespread clinical utilization of PET imaging faces significant barriers, including high operational costs, limited availability of radiopharmaceuticals, and restricted access to specialized facilities (Mac Manus & Hicks, 2010; Madison). These limitations not only constrain the medical value of PET imaging but also severely restrict the development of machine learning models, as PET datasets remain substantially smaller compared to other, often less specific but more broadly available, imaging modalities like Magnetic Resonance Imaging (MRI). Finding innovative approaches to make the benefits of PET scans more accessible, even in a simulated capacity, would significantly advance early diagnosis capabilities, enhance treatment planning protocols, and enable the development of more robust AI-based clinical tools.

Recent advances in deep learning, particularly in generative models, offer a promising avenue for simulating PET scans from readily available data. The image generation domain has grown rapidly in recent years to the point of exceptional photorealism. This rise began largely with the introduction of Generative Adversarial Networks (GANs) (Goodfellow et al., 2020; Gulrajani et al., 2017; Karras et al., 2019). GANs use adversarial training between a generator which creates data indistinguishable from real samples against a discriminator which tries to differentiate between real and generated data. GANs have gained widespread popularity for their ability to turn this simple training trick into incredibly realistic images. However, they have run into a number of challenges such as mode collapse and training instability.

So, a more direct approach which has re-popularized older score-based models to impressive success is the use of diffusion models (Ho et al., 2020; Song & Ermon, 2019; Song et al., 2020; Song & Ermon, 2020). Diffusion models iteratively refine an image out of random noise through a series of small adjustments. They are a probabilistic model which has shown the ability to produce high-quality and complex images. Furthermore, they have shown to scale effectively, with largely such models yielding incredibly realistic results (Ramesh et al., 2021; Rombach et al., 2022). Many of these diffusion model approaches have even generated conditioned images (Batzolis et al., 2021; Ramesh et al., 2021; Tashiro et al., 2021) and 3D outputs (Hoogeboom et al., 2022; Huang et al., 2023; 2024; Poole et al., 2022) that can offer insights into our task at hand.

GANs and diffusion models have also demonstrated remarkable capabilities in medical image generation (Alrashedy et al., 2022; Khader et al., 2022; Kim & Ye, 2022; Pinaya et al., 2022). Notably, conditional generation, where images are synthesized based on auxiliary information such as text or other imaging modalities, holds particular relevance (Li et al., 2019; Van den Oord et al., 2016; Yan et al., 2016). Given that PET scans are typically part of a comprehensive diagnostic imaging protocol, often following or complementing structural imaging modalities such as MRI to provide additional functional and metabolic information (Goerres et al., 2003; Meller et al., 2003), leveraging MRI as a conditioning input for PET synthesis is a logical and cost-effective approach. Potentially democratizing access to PET-like diagnostic information without incurring the substantial costs and logistical challenges associated with actual PET acquisition.

However, generating accurate and clinically relevant PET images from MRI presents several technical challenges that must be systematically addressed:

- **Data Scarcity.** The limited availability of paired MRI-PET data compared to the abundance of MRI-only data presents a fundamental constraint that necessitates techniques that can effectively utilize unpaired data.
- **3D Complexity.** Unlike conventional 2D image generation tasks, PET scans represent complex three-dimensional volumetric data with intricate spatial relationships that must be preserved throughout the generation process.
- **Structural Fidelity.** PET and MRI scans exhibit structural similarity but differ fundamentally in their functional content. Ensuring anatomical fidelity and accuracy without structural distortion during generation is crucial.

To address these challenges, we propose `MRI2PET`, a novel 3D diffusion-based method for generating PET images from MRI scans. Our approach incorporates two key innovations:

- **Style Transferred Pre-Training.** We leverage the vast amount of MRI-only data by implementing a style transfer technique that generates synthetic MRI-to-PET-like image pairs, enabling effective pre-training of the diffusion model. Style transfer is a generative technique leveraging image classification models which involves "transferring" the style of one image (e.g., color, texture) onto another while preserving the content structure. It was originally popularized in the domain of artistic image generation, but we apply it here to create "PET-like" MRI images for pre-training. This then allows the model to learn the fundamental structural relationships between MRI and PET, maximizing the utilization of available data resources while mitigating the constraints imposed by limited paired MRI-PET datasets.
- **Laplacian Pyramid Loss Component.** To enhance the fidelity and generation of important diagnostic details beyond general structural correspondence, we introduce a Laplacian pyramid loss function. This loss component ensures that the generated PET images accurately capture the multi-scale features present in real PET scans, improving diagnostic accuracy.

We evaluate `MRI2PET` using the publicly available Alzheimer's Disease Neuroimaging Initiative (ADNI) dataset, specifically on the task of generating AV45 PET scans from T1-weighted MRI imaging. While the dataset contains various types of PET images such as FDG, Amyloid, and Tau PET scans, we opt for AV45 as the most prevalent type of PET imaging within the dataset and one which under the amyloid PET classification is known to provide value in the detection and study of Alzheimer's disease as opposed to more cancer-focused modalities like FDG PET. Our comprehensive evaluation includes quantitative metrics, qualitative assessments, and clinical case studies, demonstrating that `MRI2PET` can effectively model PET scans. We further show that the generated images capture clinically relevant details, enabling their use for data augmentation and improving the performance of downstream machine learning tasks. This capability to generate

high quality, clinically relevant PET scans from MRI has the potential to expand the utility of cost-effective, automated, and accessible imaging workflows and improve the quality, efficiency, and availability of important patient diagnostics. It offers the potential to provide important PET-based diagnostic findings to more people at an earlier date to allow for earlier diagnosis and more targeted testing where relevant without waiting for the severity of the risk or condition to warrant the cost.

## 2 PROBLEM FORMULATION

We first introduce our data, problem, and task using notation utilized throughout this work.

**Definition 1 (MRI Scan Imaging)** *We denote the MRI scan data as $\mathcal{M} \in \mathbb{R}^{d_m, r_m, r_m}$, where $d_m$ represents the depth or number of slices, and $r_m$ represents the height and width resolution of the image. Each pixel or variable in $\mathcal{M}$ is normalized to lie within a defined range (we use $[0, 1]$).*

**Definition 2 (PET Scan Imaging)** *We then similarly represent the PET scan data as $\mathcal{P} \in \mathbb{R}^{d_p, r_p, r_p}$ where $d_p$ represents the PET-specific depth and $r_p$ represents the PET-specific resolution of the image. $\mathcal{P}$ is normalized to lie within a standard normal distribution for use in diffusion models, though it is subsequently converted to $[0, 1]$ for visualization.*

**Task 1 (MRI to PET Generation)** *The task is then, given an MRI scan $\mathcal{M}$, to generate the corresponding PET scan $\mathcal{P}$ for the same patient such that it closely mirrors the true PET results. This is achieved by learning and sampling from $P(\mathcal{P}|\mathcal{M})$.*

## 3 MRI2PET METHOD

Our proposed MRI2PET method, illustrated in Figure 1, employs a conditional diffusion modeling framework enhanced by two critical innovations designed to address key challenges in synthesizing PET images from MRI scans. These contributions are as follows:

(1) **Style Transferred Pre-Training** that makes use of the wealth of MRI images as compared to the small number of PET scans by using the unpaired MRIs for pre-training. However, to better simulate our final task, we use style transfer to convert them to PET-like MRI images for this stage.

(2) **Laplacian Pyramid Loss Component** that emphasizes the need to capture important details, beyond simple structural patterns, which are crucial to differentiate outcomes and provide downstream utility. It does this using Laplacian pyramids which consist of image details at various resolutions as calculated by the difference between true images and those which are downsampled then upsampled. To enforce the learning of these details, we include an objective of similar Laplacian pyramids between real and generated PET images.

We now proceed through the diffusion framework as well as these contributions in more detail.

### 3.1 (I) 3D DIFFUSION FRAMEWORK

We use a standard diffusion process to transform a sample from a simple Gaussian source distribution to one from the complex PET scan target distribution via a series of small diffusive adjustments.

**Iterative Diffusion** The forward noising pathway consists of repeatedly adding noise to an initial PET image until it resembles a fully noisy sample from our source Gaussian distribution. The PET image is then recovered by reversing the process and denoising the noisy sample step-by-step. Formally, the noising process at diffusion step $t$ is represented by:

$$d\hat{\mathcal{P}}^{(t)} = \sqrt{\hat{\alpha}_t}\hat{\mathcal{P}}^{(t)}dt + \sqrt{1 - \hat{\alpha}_t}dW^{(t)} \tag{1}$$

where $\hat{\mathcal{P}}^{(t)}$ denotes the noisy PET image at step $t$, $\sqrt{\alpha_{\hat{t}}}$ is the drift term that scales the image intensity, $\sqrt{1 - \alpha_{\hat{t}}}$ is the diffusion term that scales the noise, and $dW^{(t)}$ is an infinitesimal Wiener process representing the random noise $\varepsilon$. We discretize this continuous diffusion over $T = 1000$ noise steps, and we set $\hat{\alpha}_t$ by defining $\beta$ as a 1000-step linear spacing between $\beta_{\text{start}} = 0.0015$ and $\beta_{\text{end}} = 0.02$ with $\alpha = 1 - \beta$ and $\hat{\alpha}_t$ the cumulative product of the first $t$ values of $\alpha$.

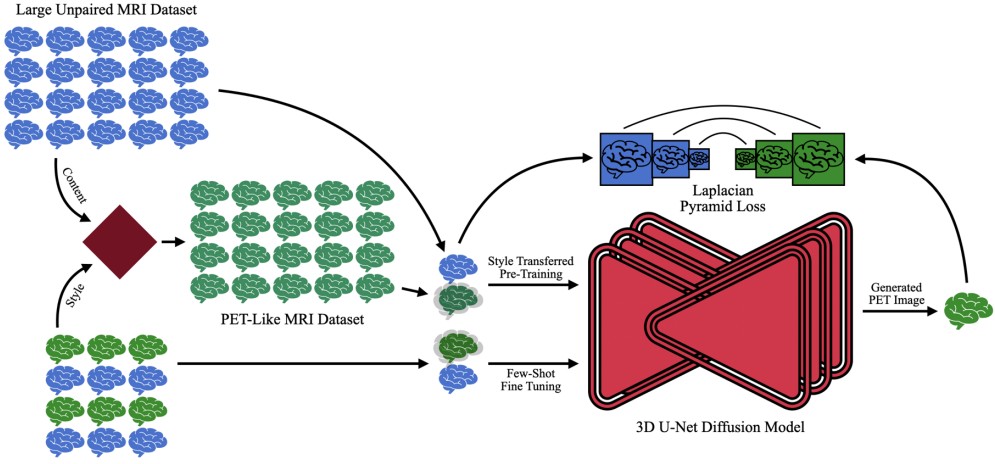

Figure 1: Schematic illustration of the `MRI2PET` architecture. The method employs a 3D U-Net diffusion model initially pre-trained on a large dataset of unpaired MRI images. Style transfer techniques are utilized to generate PET-like MRI images, effectively simulating MRI-to-PET conditions for pre-training. The model is subsequently fine-tuned on a smaller, paired MRI-PET dataset using a Laplacian pyramid loss, emphasizing the preservation and enhancement of critical multi-scale image details essential for clinical diagnostics.

**Denoising with DDPM** We conduct the reverse denoising process using a neural network which serves as our denoising diffusion probabilistic model (DDPM). This model directs the denoising process at each diffusion step given the noisy image and conditioning information (in our case, the time step number and MRI image). It does this by predicting the error $\epsilon$ between the provided noisy image $\hat{\mathcal{P}}^{(t)}$ and the true target image $\mathcal{P}$. The generation process then takes an input image initialized as pure Gaussian noise and iteratively feeds that image along with the corresponding MRI to the trained model to repeatedly refine the output and generate realistic PET scan results.

We use a UNet autoencoder as our DDPM, as is standard within the domain for use with diffusion models Ronneberger et al. (2015). UNet architectures consist of an encoding pathway, which embeds the input image into lower-dimensional space, and a subsequent decoding pathway which maps that embedding back into the original dimensionality. Each step in either pathway is furthermore connected via a skip connection to maintain spatial information as well as the relevant conditioning information. The only adaptations we make to the architecture here are replacing the standard 2D convolutions with 3D versions (representing our PET images as a single-channel 3D structure rather than a multi-channel 2D structure) and replacing the standard complete attention with a linear version in order to reduce the computational complexity to accommodate our computing resources. Further details including the number of blocks in either pathway and dimensionality throughout are provided in our supplement.

## 3.2 (II) STYLE TRANSFERRED PRE-TRAINING

One of the significant challenges in generating PET images from MRI scans is the limited availability of paired MRI-PET datasets. Due to their cost and specificity, PET scan images are much rarer than MRIs. So, to address the limited amount of image pairs, we leverage the abundant unpaired MRI images for pre-training our model. However, our core task is a conditional generation task going from MRI to PET images. So, we need to convert isolated MRI images into a conditional setup. The obvious solution is to condition an MRI on itself (possibly with noise to prevent the task from devolving into pure copying). However, we take this one step further to simulate our desired task as closely as possible and ensure that the model learns features relevant to PET imaging. Specifically, we employ style transfer techniques to convert these unpaired MRI images into PET-like MRI images.

We do this using a pre-trained VGG19 model to adapt the style (such as intensity patterns and noise characteristics) of a randomly selected PET image within our dataset while preserving the anatomical structure of the MRI. Since the pre-trained model only accepts 2D inputs, we do this one slice at a time iteratively over the depth dimension of the content MRI style PET pair. This process enables the `MRI2PET` model to be pre-trained on a large dataset that closely simulates the MRI to PET generation task by training on these MRI to PET-like MRI image pairs. As such, the model learns to recognize and generate features that are relevant to PET scans, even before being exposed to actual paired MRI-PET data, significantly enhancing the model's ability to generate realistic PET images once fine-tuning on the limited paired dataset begins.

### 3.3 (III) LAPLACIAN PYRAMID LOSS COMPONENT

Finally, we add an additional loss component onto the standard mean squared error based training objective which instructs the DDPM to accurately predict the noise. This additional objective aims to enhance the model's ability to capture important details at multiple resolutions within the generated images. It does so by leveraging the Laplacian pyramid, a multi-scale image representation consisting of a set of residuals between increasingly blurred and lower-dimensional images.

Specifically, a Laplacian pyramid is constructed by first creating a Gaussian pyramid by iteratively upsampling (through a 2x2 averaging pooling operation) the original image. The Laplacian pyramid then takes each layer of the Gaussian pyramid and compares it to the layer below, upsampled via bilinear interpolation back to the same size. So, it effectively compares the original image, scaled to different, increasingly small resolutions, with blurrier versions of itself. The differences in these comparisons are interpreted to be the details at a given resolution.

We then use the Laplacian pyramid to construct an additional loss component as follows:

1. Create a "predicted image" using the noise predictions of the DDPM along with its noised input.
2. Construct 5-step Laplacian pyramids for both the generated and target images
3. For each level in the pyramid, compute the mean squared error (MSE) between the corresponding levels of the generated and target images.
4. Sum the MSE values across all levels to obtain the final Laplacian pyramid loss.

which can be formalized by

$$\mathcal{L}_{\text{Lap}}(\mathcal{P}, \hat{\mathcal{P}}) = \sum_{l=1}^{5} \text{MSE}(\text{LapPyr}(\mathcal{P})_l, \text{LapPyr}(\hat{\mathcal{P}})_l) \tag{2}$$

where a given Laplacian Pyramid is calculated by

$$\text{LapPyr}(P) = \{P_l - \text{Upsample}(\text{Downsample}(P_l))\}_{l=1}^{5} \tag{3}$$

The total loss function for training the DDPM is then defined as a weighted combination of the standard noise prediction loss and this Laplacian pyramid loss:

$$\mathcal{L} = \text{MSE}(\epsilon, \tilde{\epsilon}) + \lambda \mathcal{L}_{\text{Lap}}(\mathcal{P}, \tilde{\mathcal{P}}) \tag{4}$$

where $\epsilon$ is the noise added to $\mathcal{P}$ to produce the model input $\hat{\mathcal{P}}$, $\tilde{\epsilon}$ is the DDPM model's prediction, $\tilde{\mathcal{P}}$ is the predicted image based on that predicted noise as calculated by $\tilde{\mathcal{P}} = \frac{1}{\sqrt{\bar{\alpha}}} \cdot \hat{\mathcal{P}} - \frac{\sqrt{1-\bar{\alpha}}}{\bar{\alpha}} \cdot \tilde{\epsilon}$, and $\lambda$ is the weighting factor, which we experimentally set to 0.25. This composite loss encourages the model to not only minimize the pixel-wise differences but also to preserve structural and textural details across multiple scales, leading to more accurate, detailed, and visually coherent image generation.

## 4 EXPERIMENTS

### 4.1 EXPERIMENTAL DESIGN

We rigorously evaluate `MRI2PET` using the publicly available ADNI dataset (Mueller et al., 2005). Our experimental framework addresses three primary research questions:

1. Can `MRI2PET` generate realistic PET images from MRI?

Table 1: Quantitative Generation Quality

|  | FID ($\downarrow$) | SSIM ($\uparrow$) | PSNR ($\uparrow$) |
|---|---|---|---|
| GAN | $303.615 \pm 0.063$ | $0.101 \pm 0.002$ | $13.712 \pm 0.025$ |
| Vanilla Diffusion | $63.844 \pm 0.046$ | $0.764 \pm 0.005$ | $21.005 \pm 0.137$ |
| CDC | $287.531 \pm 0.067$ | $0.021 \pm 0.000$ | $7.780 \pm 0.040$ |
| DCL | $269.323 \pm 0.045$ | $0.076 \pm 0.002$ | $0.958 \pm 0.015$ |
| DiffAugment | $309.005 \pm 0.052$ | $0.055 \pm 0.001$ | $7.901 \pm 0.028$ |
| MaskGAN | $302.071 \pm 0.048$ | $0.016 \pm 0.000$ | $4.629 \pm 0.020$ |
| DDPM-PA | $265.000 \pm 0.051$ | $0.035 \pm 0.000$ | $8.108 \pm 0.041$ |
| w/o Style Transfer | $62.569 \pm 0.025$ | $0.854 \pm 0.004$ | $24.615 \pm 0.130$ |
| w/o Pre-Training | $108.401 \pm 0.046$ | $0.260 \pm 0.008$ | $15.644 \pm 0.134$ |
| w/o Loss | $\mathbf{61.213 \pm 0.029}$ | $0.827 \pm 0.004$ | $23.156 \pm 0.120$ |
| MRI2PET | $61.735 \pm 0.032$[***] | $\mathbf{0.857 \pm 0.004}$[***] | $\mathbf{24.771 \pm 0.123}$[***] |

2. Do synthetic PET images generated by `MRI2PET` replicate clinically meaningful patterns observed in actual patient PET images?
3. Does augmenting datasets with `MRI2PET`-generated PET images enhance performance in downstream machine learning tasks?

## 4.2 DATASET

We utilized publicly accessible paired MRI-PET scans from the Alzheimer's Disease Neuroimaging Initiative (ADNI) dataset, which included 2,492 image pairs defined as scans from the same patient captured within one year (Mueller et al., 2005). Additionally, we incorporated 22,956 unpaired MRI images drawn from ADNI, UK Biobank (Sudlow et al., 2015), and Parkinson's Progression Markers Initiative (PPMI) datasets (Marek et al., 2018).

All MRI scans underwent standard preprocessing steps, including template registration, skull stripping, and bias correction. MRI images were then resampled to a 64×144×144 voxel dimensionality using windowed sinc interpolation and normalized to a $[0, 1]$ intensity range.

PET images required handling temporal sequences, where applicable. Each frame was rigidly registered first to the initial frame, and subsequently to the temporal mean. The final PET image was the temporal mean, which was affine-registered to its corresponding MRI, resampled to a 32×128×128 voxel dimensionality via windowed sinc interpolation, and normalized.

For pre-training involving MRI outputs, MRI images were further resampled to PET dimensions (32×128×128) using quintic interpolation. We split the paired dataset randomly, reserving 20% for testing. The remaining paired data, combined with unpaired MRI scans, were further split into training (90%) and validation (10%) sets.

We pre-trained `MRI2PET` on the unpaired MRI dataset for 250 epochs, then fine-tuned on the paired MRI-PET dataset for 5,000 epochs. Training utilized the PyTorch framework (Paszke et al., 2019), with a batch size of 128, a learning rate of 0.0001, and the Adam optimizer.

## 4.3 BASELINES

We benchmark `MRI2PET` against a suite of baselines to comprehensively validate performance:

**External Baselines:**

1. **GAN**: Conditional Wasserstein GAN (Gulrajani et al., 2017) trained on the limited paired data.
2. **Vanilla Diffusion**: Standard conditional diffusion model trained on limited paired data.
3. **CDC**: Few-shot GAN with cross-domain correspondences. (Ojha et al., 2021)
4. **DCL**: Few-shot GAN optimizing diversity. (Zhao et al., 2022)
5. **DiffAugment** Differentiable augmentation GAN. (Zhao et al., 2020)
6. **MaskGAN**: GAN with masked discriminator features. Zhu et al. (2024)
7. **DDPM-PA**: Few-shot diffusion model with pairwise adaptation. Zhu et al. (2022)

**Ablation Studies:**

1. **MRI2PET w/o Style Transfer**: MRI-to-MRI noise-conditioned pre-training without style transfer.

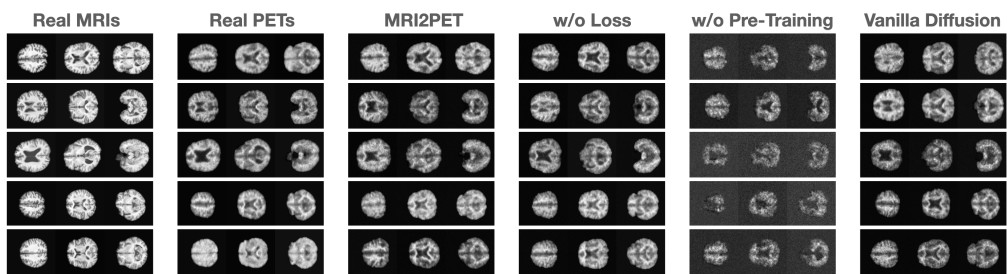

Figure 2: Qualitative comparison of axial brain slices from five randomly selected test patients

2. **MRI2PET w/o Pre-Training**: Direct training without pre-training.
3. **MRI2PET w/o Loss**: Training without additional Laplacian pyramid loss.

These rigorous comparative analyses were designed to highlight the contribution of each methodological innovation and validate MRI2PET's overall effectiveness.

### 4.4 MRI2PET GENERATES REALISTIC PET IMAGES FROM MRI SCANS

The primary aim of MRI2PET is the generation of realistic PET images from MRI data. We quantitatively assessed image quality using three established metrics: Fréchet Inception Distance (FID), Structural Similarity Index Measure (SSIM), and Peak Signal-to-Noise Ratio (PSNR). FID evaluates the similarity between distributions of real and synthetic images based on features extracted by a pre-trained Inception v3 model. Given that Inception v3 requires 2D inputs, we performed axial slicing of our 3D datasets. SSIM and PSNR were computed in 3D to evaluate structural detail preservation and noise levels, respectively. Results are summarized in Table 1 averaged over all of the images in our held-off test dataset.

MRI2PET significantly outperformed baseline methods across all metrics, particularly in SSIM and PSNR, indicating superior structural preservation, reduced noise, and enhanced visual quality in the generated PET images. While FID scores between MRI2PET and Vanilla Diffusion were similar, MRI2PET showed clear superiority over other baselines, with a notable improvement of more than 12% in SSIM and PSNR compared to the next-best external baseline. Ablation studies further clarified the roles of the individual contributions: removal of style-transferred pre-training significantly degraded performance, confirming its importance for effectively utilizing unpaired MRI data. Similarly, omitting Laplacian pyramid loss negatively impacted the quality of generated images, underscoring its crucial role in capturing detailed, multi-scale image features essential for clinical diagnostic accuracy. Collectively, these findings highlight the complementary strengths of these enhancements and their combined necessity for optimal performance.

Qualitative evaluations of axial slices from randomly selected test patients further reinforced these findings (Figure 2). MRI2PET consistently preserved patient-specific anatomical details such as brain size, shape, and orientation, closely mirroring real PET images and outperforming baselines.

Notably, structural intricacies, including distinct anatomical variations such as larger or smaller brain sizes and unique shape characteristics (e.g., horizontally elongated brains), were effectively captured. Moreover, MRI2PET reliably aligned physiological details between MRI and PET slices, demonstrating precise vertical alignment across modalities, which further validates the anatomical accuracy of generated images. These results provided a stark visual contrast compared to baseline methods, particularly highlighting MRI2PET's substantial improvements over GAN-based and diffusion-based methods that consistently struggled with the complexities inherent in 3D image generation tasks.

### 4.5 MRI2PET CAPTURES CLINICALLY RELEVANT PATTERNS

To examine MRI2PET's ability to capture clinically meaningful patterns, we conducted detailed case-study analyses. In patients exhibiting significant brain atrophy (Figure 3a), MRI2PET-generated PET images accurately reflected regions of decay observed in corresponding MRIs,

**Atrophy**

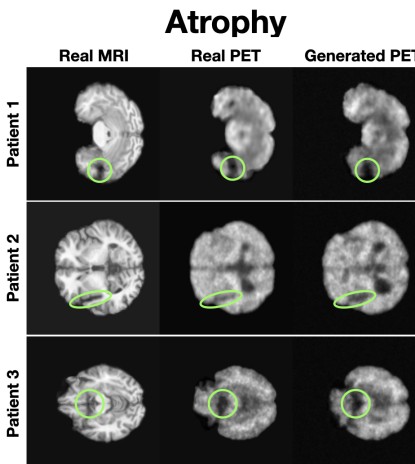

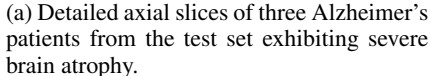

**Longitudinal AD Progression**

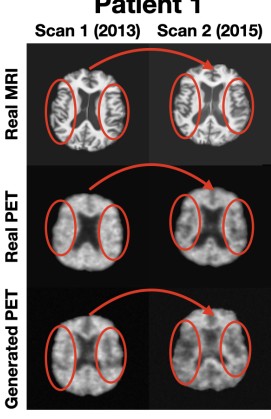

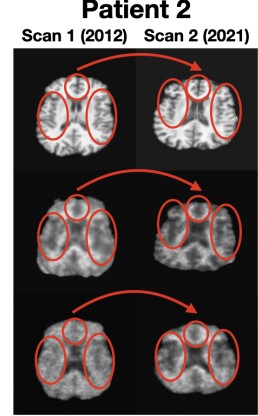

(a) Detailed axial slices of three Alzheimer's patients from the test set exhibiting severe brain atrophy.

(b) Two distinct test patients with two scans each, illustrating longitudinal Alzheimer's progression with increased brain atrophy in the second set of imaging.

Figure 3: Case studies depicting (a) severe brain atrophy in three patients and (b) longitudinal Alzheimer's progression across two patients.

demonstrating sensitivity to disease-specific anatomical changes. For instance, specific areas of notable atrophy, such as reduced cortical thickness or enlarged ventricles, were reliably replicated in the synthetic PET images, highlighting `MRI2PET`'s precision in capturing disease-associated morphological alterations. This speaks to the patient-specific and clinically relevant quality of `MRI2PET`'s generations, even as the atrophy itself may not be central to the downstream utility and usage of the PET imaging.

In longitudinal case studies involving Alzheimer's disease progression (Figure 3b), `MRI2PET` effectively captured deterioration patterns over time. Notably, increased prominence and depth of temporal sulci—a hallmark feature of progressive Alzheimer's disease—were accurately reflected in sequential PET scans generated by `MRI2PET`. This temporal sensitivity to progressive anatomical degradation underscores `MRI2PET`'s potential utility for monitoring disease progression, facilitating timely interventions, and enabling more precise longitudinal clinical evaluations. These case studies collectively confirm `MRI2PET`'s robust capability in replicating disease-specific temporal and anatomical patterns, significantly contributing to its potential clinical utility in diagnostics and patient monitoring.

## 4.6 `MRI2PET` ENHANCES DOWNSTREAM MACHINE LEARNING TASKS

We further assessed `MRI2PET`'s practical utility beyond visual realism by examining its impact on downstream machine learning tasks, specifically Alzheimer's disease classification and Mini-Mental State Examination (MMSE) score prediction, leveraging clinically relevant patient labels from the ADNI dataset. For Alzheimer's disease classification, we utilized the ADNI dataset's disease labels categorizing patients into cognitively normal, mildly cognitively impaired, or Alzheimer's disease groups as a multi-class classification problem. The MMSE task employed scores from a standard cognitive examination, restricted to scores of 20 or above (out of a maximum of 30) to maintain focus on patients with mild or no cognitive impairment. Information regarding these MMSE distributions can be found in the demographics tables in the supplemental material. We systematically evaluated three model types trained with distinct raw imaging inputs: MRI-only, PET-only, and MRI-PET paired data. Training involved multiple data configurations to comprehensively assess `MRI2PET`'s synthetic image utility:

- **Real MRI:** Large, unpaired MRI dataset.
- **Real Limited MRI:** Limited MRI data from paired MRI-PET scans.
- **Real PET:** Limited PET data without MRI pairing.

Table 2: Performance results of downstream Alzheimer's Disease classification tasks

|  | Accuracy (↑) | F1 Score (↑) | Balanced Accuracy (↑) | AUROC (↑) |
|---|---|---|---|---|
| Real MRI | $0.520 \pm 0.014$ | $0.497 \pm 0.016$ | $0.506 \pm 0.014$ | $0.704 \pm 0.012$ |
| Real Limited MRI | $0.449 \pm 0.009$ | $0.369 \pm 0.020$ | $0.401 \pm 0.012$ | $0.611 \pm 0.014$ |
| Real PET | $0.542 \pm 0.009$ | $0.473 \pm 0.020$ | $0.498 \pm 0.013$ | $0.688 \pm 0.014$ |
| MRI2PET Generated PET | $0.542 \pm 0.009$ | $0.468 \pm 0.019$ | $0.496 \pm 0.012$ | $0.685 \pm 0.012$ |
| Real Paired | $0.559 \pm 0.006$ | $0.518 \pm 0.006$ | $0.526 \pm 0.007$ | $0.723 \pm 0.004$ |
| MRI2PET Augmented Paired | $\mathbf{0.605 \pm 0.007}^{***}$ | $\mathbf{0.595 \pm 0.008}^{***}$ | $\mathbf{0.594 \pm 0.008}^{***}$ | $\mathbf{0.780 \pm 0.005}^{***}$ |

Table 3: Performance results of downstream Alzheimer's Disease classification tasks

|  | MSE (↓) | Pearson Correlation (↑) |
|---|---|---|
| Real MRI | $5.538 \pm 0.148$ | $0.356 \pm 0.034$ |
| Real Limited MRI | $6.271 \pm 0.027$ | $0.110 \pm 0.014$ |
| Real PET | $5.880 \pm 0.113$ | $0.225 \pm 0.035$ |
| MRI2PET Generated PET | $6.148 \pm 0.066$ | $0.148 \pm 0.026$ |
| Real Paired | $5.601 \pm 0.133$ | $0.294 \pm 0.038$ |
| MRI2PET Augmented Paired | $\mathbf{4.982 \pm 0.116}^{**}$ | $\mathbf{0.458 \pm 0.024}^{***}$ |

- **Synthetic PET:** Limited synthetic PET images generated by MRI2PET.
- **Real Paired:** Limited real MRI-PET pairs.
- **Augmented Paired:** Real MRI-PET pairs augmented by additional synthetic PET images paired with previously unpaired MRIs.

Performance across these datasets is detailed in Table 2 (classification) and Table 3 (MMSE regression) where we train 25 different models on the original training and validation datasets (the same as MRI2PET was trained on) using different random seeds for each task-model type pair and report the mean metrics results along with the standard errors over those runs, calculated on the test dataset (which is paired and so can be used for any type of input). The synthetic PET dataset alone closely approximated results achieved with real PET data, confirming MRI2PET's capability to generate clinically informative synthetic scans. Crucially, training with augmented paired datasets resulted in the highest performance improvements, surpassing both real paired and individual modality datasets across accuracy, F1 score, balanced accuracy, AUROC (classification), and mean squared error, and Pearson correlation (MMSE regression). Specifically, the augmented paired dataset achieved classification accuracy of $0.605 \pm 0.007$ and F1 scores of $0.595 \pm 0.008$, representing substantial improvements over all other configurations. Similarly, MMSE prediction with augmented data demonstrated significantly reduced error ($4.982 \pm 0.116$ MSE) and increased correlation ($0.458 \pm 0.024$), clearly outperforming alternative approaches. These findings underscore MRI2PET's capability to effectively expand available imaging data, significantly enhancing the robustness, accuracy, and predictive power of downstream machine learning models, demonstrating real-world clinical utility beyond simple data augmentation. Finally, examining the MMSE predictions offered additional validation. Using a regression model trained exclusively on real PET images, we compared predictions for real and MRI2PET-generated PET scans, revealing a notable positive correlation (r = 0.273). This further affirms MRI2PET 's ability to capture diagnostically relevant information within generated PET scans.

## 5 CONCLUSION

In this study, we introduced MRI2PET, an innovative diffusion-based generative method designed to simulate PET scan imaging from MRI data. MRI2PET specifically addresses significant technical challenges—including the scarcity of paired MRI-PET data, the inherent complexity of 3D volumetric data, and the structural-functional relationships between MRI and PET modalities—by integrating two novel components: a style-transfer-based pre-training strategy and a Laplacian pyramid training objective. Through extensive evaluation using the Alzheimer's Disease Neuroimaging Initiative (ADNI) dataset, MRI2PET consistently outperformed various generative baselines, demonstrating superior quantitative performance metrics (FID, SSIM, PSNR) and producing qualitatively realistic PET images that closely reflect patient-specific anatomical details.

Beyond the general fidelity and realism of the generated PET images, `MRI2PET` effectively captured clinically meaningful details, such as brain atrophy and specific patterns associated with Alzheimer's disease progression. This capability significantly enhanced its practical utility in downstream machine learning applications. Models trained on datasets augmented by synthetic PET images from `MRI2PET` substantially outperformed those trained solely on real datasets, underscoring the method's potential to effectively compensate for missing PET imaging data and thereby boost diagnostic and predictive modeling performance.

So, `MRI2PET` represents a promising and clinically relevant advancement toward simulating PET imaging outcomes from MRI data, delivering immediate practical utility and opening avenues for further methodological refinements and enhanced interpretability in a variety of medical settings.

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

Table 4: Number of MRI Images by dataset

| Total | ADNI | PPMI | UKBiobank |
|---|---|---|---|
| 22,956 | 11,453 | 1,956 | 9,547 |

| | Train | Validation | Test |
|---|---|---|---|
| Number of PET Scans | 2,017 | 225 | 250 |
| Number of Patients | 1,068 | 205 | 232 |
| Percent Male | 51.2% | 55.9% | 48.7% |
| Average Age at Intake | $71.93 \pm 7.12$ | $71.36 \pm 6.40$ | $71.06 \pm 6.66$ |
| Number of CN | 393 | 89 | 98 |
| Number of MCI | 355 | 63 | 67 |
| Number of AD | 306 | 52 | 66 |
| Average MMSE | $25.75 \pm 5.03$ | $26.37 \pm 4.55$ | $25.62 \pm 5.56$ |
| Number MMSE 20+ | 905 | 182 | 195 |
| Average MMSE 20+ | $27.16 \pm 2.75$ | $27.46 \pm 2.45$ | $27.28 \pm 2.76$ |

Table 5: Dataset Statistics and Demographics of the ADNI PET Dataset

## A  EXPERIMENTAL DETAILS

To conduct our experiments, we train each compared method and generate PET scan images based on each MRI scan in our test dataset. We also generate PET results based on all MRI images from the ADNI dataset in our other splits with just our MRI2PET model. We then perform a variety of analyses comparing the similarity in terms of quantitative correspondence, visual adherence, and downstream utility of these generated PET scans to the true PET results. We report 95% confidence intervals for each experimental value for each metric, calculated by performing 100 bootstraps of the test set. We present the best values in bold for each metric, and we bold any other values whose means fall into that best value's confidence interval. We also provide asterisks along with each tabular result, referring to p-value significance based on a two-tailed t-test comparing augmented results to the nearest baseline approach with $*$ signifying $p < 0.05$, $**$ signifying $p < 0.01$, and $***$ signifying $p < 0.001$. We perform all experimentation using Nvidia A100 GPUs on a large computing cluster.

## B  DATASET STATISTICS

To provide more detailed information regarding our datasets and offer context for our experiments, we provide more detailed count and demographic information in a pair of tables. First, we provide a count of where the MRI images were found across our three datasets in Table 4. Then, we provide a variety of demographics and counts, especially pertaining to our downstream experiments and their labels in Table **??**.

## C  A POSSIBLE BIOLOGICAL EXPLANATION FOR GENERATIVE QUALITY

We conclude with an intriguing qualitative observation and a hypothesis which bridges the biological and generative mechanisms regarding variations in generative image quality across different patient samples. Synthetic PET images generated by MRI2PET exhibited notably greater clarity and anatomical accuracy in patients presenting clear structural brain atrophy on MRI scans, in contrast to cases without evident MRI-detectable decay. Further analysis revealed a consistent pattern: PET images generated for patients at later stages of Alzheimer's disease (characterized by pronounced atrophy visible on MRI) were significantly more precise and detailed than those at earlier stages without obvious structural changes, a pattern demonstrated in Figure 4.

To explain this observation, we proposed a biological hypothesis rooted in the known pathological progression of Alzheimer's disease, particularly involving amyloid-beta (A$\beta$) plaque accumulation.

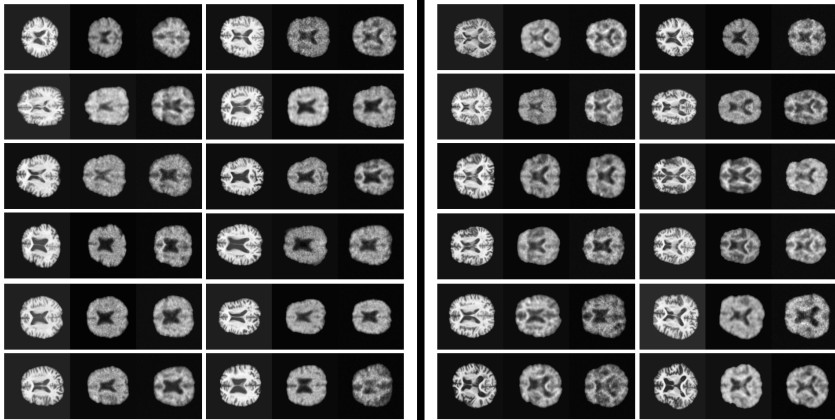

Figure 4: A large number of images for Alzheimer's patients throughout the test dataset. Each triplet has the real MRI on the left, real PET in the middle, and generated PET on the right. They are broadly sorted into patients with clear brain decay based on the MRI on the right half of the image and patients without clear decay on the left. Notably, the generated image quality on the left is less sharp as regardless of the internal content or quality of the real MRI and PET images, those images have a general blurriness, though there are exceptions in quality on either side.

Amyloid-beta plaques typically accumulate prior to the structural atrophy visible through MRI, which is itself driven by neuronal loss and tau disease pathology. Thus, patients at early disease stages—who might exhibit substantial underlying amyloid pathology yet no visible structural MRI abnormalities—can present multiple plausible metabolic patterns detectable by PET. The generative uncertainty associated with these multiple potential outcomes may lead MRI2PET to produce less distinct, blurred synthetic images as the model implicitly averages across these scenarios rather than clearly delineating one distinct outcome.

Although this averaging behavior is suboptimal from a purely generative modeling perspective, where distinct scenario generation would be preferable, it significantly highlights MRI2PET 's responsiveness to underlying biological processes. Indeed, this phenomenon suggests that MRI2PET is implicitly sensitive to biologically meaningful variations related to early Alzheimer's pathology that structural MRI alone cannot reveal. Recognizing this biological underpinning provides valuable insights into possible avenues for future model refinement, such as explicitly modeling uncertainty or incorporating early-stage biomarkers.

Therefore, while the observed variability in generative quality indicates opportunities for technical improvement, it also underscores MRI2PET 's potential utility as a tool for capturing subtle pathological insights, potentially facilitating earlier diagnosis and more precise differentiation among various Alzheimer's disease stages.

It also underscores important limitations to be aware of within the core task setup. Generating PET images based on MRI scan results can by definition inject no additional information than is already available within that MRI. This does not reduce the value or potential applications. There may be lots of information that is not readily discerned from looking at the MRI itself; we have shown in this paper that we succeed in injecting utility through the augmentation of MRI-PET datasets, and the ability to simulate a specific instantiation within the range of potential corresponding outcomes, especially a physiologically meaningful one, offers clearly valuable applications. However, it does mean that this value and these applications should be conditioned, and this setup should be noted.

