# OpenReview forum: "MRI2PET: Realistic PET Image Generation from MRI for Automated Inference of Brain Atrophy and Alzheimer’s"
_ICLR.cc/2026/Conference — ICLR 2026 Conference Withdrawn Submission_

### Official Review · Reviewer_Gf63 · 2025-10-24

**Soundness:** 1
**Presentation:** 2
**Contribution:** 1
**Rating:** 2
**Confidence:** 4

**Summary:**

The paper proposes using a Laplacian pyramid loss and pseudo-data pretraining to improve MRI-to-PET translation.

The problem itself is important and has a long history in multimodal medical imaging and computer vision, but this paper does not acknowledge prior work or provide a comprehensive literature review. In addition, the baseline methods used in this study perform poorly (and confusingly, the DDPM method performs worse than the vanilla diffusion model). The results are also contradictory—specifically, the proposed method without pretraining performs worse than vanilla diffusion, suggesting that adding the Laplacian loss provides little to no benefit. Furthermore, the visualizations of the results do not demonstrate clinical relevance—the synthetic images exhibit artifacts that could potentially lead to misdiagnosis.

**Strengths:**

1. The two methods proposed are in themselves interesting: The Laplacian pyramid loss encourages focusing on low spatial frequency components and the pseudo data for pre-training also helps incorporate some prior knowledge about the PET distribution.

**Weaknesses:**

**[W1] **Concerns Regarding Quantitative Results (Table 1)****

W1-1. **Discrepancy with Established Baselines**: The performance metrics reported in Table 1 for baseline methods (like GANs and vanilla diffusion) appear surprisingly low compared to results published in existing literature. For instance, recent works [1, 2] often show much stronger baseline performance (e.g., Pix2pix achieving 93 SSIM and 73 PSNR in [1]). This significant discrepancy raises concerns about the experimental setup or the reported metrics.

W1-2. **Counter-intuitive Ablation Results**: The results for the proposed method's components are also confusing. It is unclear why advanced diffusion variants like diffAugment or DDPM-PA, which supposedly improve upon vanilla diffusion, are reported as performing worse than simpler baselines.

W1-3. **Potential Self-Contradiction**: There appears to be an internal contradiction in Table 1. The vanilla diffusion model is listed with an FID of 63, while the proposed MRI2PET w/o pretraining (the model without the key contribution) scores 108. This suggests that the authors' additions of style transfer and the Laplacian loss significantly degrade performance compared to a standard baseline. Is there another explanation?

**[W2] Clarity Needed on Dataset Splitting**

A critical detail is missing from the experimental description (Lines 302-305): Was the dataset split by patient (subject) or by slice? If the split was by slice, images from the same patient could exist in both the training and test sets, leading to data leakage and an overestimation of the model's true generalization performance.

**[W3]  Concerns Regarding Qualitative Results (Figure 3)**

The qualitative examples in Figure 3-b seem to highlight significant limitations of the model, which undermines confidence in its utility. In "Patient 1 - Scan 1," the real PET image shows normal intensity in the temporal lobe. However, the generated PET image shows decreased PET intensity in that exact region. This is a clinically significant error. A synthetic PET image with such an artifact could lead a clinician to a misdiagnosis. A similar issue appears to be present in the images for "Patient 2."

**[W4] No literature review**

The authors seem to claim that the main contribution is proposing a new task of MRI-to-PET, and this can be confusing and misleading. There are plenty of work in this domain such as reviewed in [3].

[1] MRI-to-PET Cross-Modality Translation using Globally & Locally Aware GAN (GLA-GAN) for Multi-Modal Diagnosis of Alzheimer’s Disease

[2] High-quality PET image synthesis from ultra-low-dose PET/MRI using bi-task deep learning

[3] Cross-modality Neuroimage Synthesis: A Survey

**Questions:**

1. Can the authors explain why the performance of baselines are so bad in Table 1? What are the main reasons that it is far below previous work.

2. Can the author add more details about dataset splitting?

3. Would the authors add literature review for the work and highlight the main difference?

---

### Official Review · Reviewer_uNvu · 2025-10-30

**Soundness:** 2
**Presentation:** 3
**Contribution:** 2
**Rating:** 2
**Confidence:** 4

**Summary:**

This paper proposes a 3D diffusion-based model that generates AV45-PET scans from T1-weighted MRI images. This method incorporates style-transferred pre-training and a Laplacian pyramid loss to leverage unpaired MRI data and structural correspondences between modalities while simultaneously emphasizing the crucial details.

**Strengths:**

(1) The proposed method leverages the vast amount of MRI-only data by implementing a style transfer technique that generates synthetic MRI-to-PET-like image pairs, enabling effective pre-training of the diffusion model.

(2) This paper introduces a Laplacian pyramid loss function to enhance the fidelity and generation of important diagnostic details beyond general structural correspondence.

(3) Experimental results show the effectiveness of the proposed framework.

**Weaknesses:**

(1) The paper leverages the vast amount of MRI-only data for pre-training and implements a style transfer technique that generates synthetic MRI-to-PET-like image pairs. While these technologies are commonly used in existing pre-training and image translation works, the overall novelty is limited.

(2) The comparison baselines are not sufficient, and more state-of-the-art synthesis methods should be included in the comparison experiments.

(3) The effectiveness of the proposed framework is not validated on one dataset (ADNI).

**Questions:**

(1) The paper leverages the vast amount of MRI-only data for pre-training and implements a style transfer technique that generates synthetic MRI-to-PET-like image pairs. While these technologies are commonly used in existing pre-training and image translation works, the overall novelty is limited.

(2) The comparison baselines are not sufficient, and more state-of-the-art synthesis methods should be included in the comparison experiments.

(3) The effectiveness of the proposed framework is not validated on one dataset (ADNI).

---

### Official Review · Reviewer_p3aj · 2025-11-11

**Soundness:** 2
**Presentation:** 2
**Contribution:** 2
**Rating:** 4
**Confidence:** 3

**Summary:**

This paper proposes MRI2PET, a 3D diffusion-based generative model that synthesizes AV45-PET scans from T1-weighted MRI images. The model is evaluated using the ADNI dataset and compared against multiple GAN- and diffusion-based baselines. Results show improvements in quantitative metrics (FID, SSIM, PSNR), qualitative realism, and downstream clinical tasks (classification and MMSE regression).

**Strengths:**

1)Comprehensive framework including Integrates unpaired data utilization, multi-scale loss, and clinical validation within a single pipeline.
2)The proposed approach outperforms several competitive baselines and improves clinical downstream tasks.
3)The discussion on Alzheimer’s pathology and MRI–PET relationships provides valuable translational insight.

**Weaknesses:**

1)The use of diffusion models, Laplacian losses, and style-transfer pre-training has precedents in recent literature; their combination, while practical, is incremental.
2)All experiments rely on ADNI; results on independent datasets (e.g., UK Biobank, PPMI PET) would strengthen claims of generalizability.
3)Since MRI does not uniquely determine PET uptake, uncertainty-aware or probabilistic evaluation would be informative.

**Questions:**

1)How sensitive is the model to the style-transfer step? Would using a learned generative PET-like mapper outperform classical VGG19-based transfer?
2)How does MRI2PET perform on other PET modalities (e.g., FDG, Tau)? Could the method generalize beyond AV45?
3)What is the computational cost (training/inference time) and feasibility for clinical deployment?
4)Could MRI2PET’s output be used to infer uncertainty or confidence intervals in generated PET intensity?

---

### Official Review · Reviewer_qyDs · 2025-11-12

**Soundness:** 3
**Presentation:** 2
**Contribution:** 2
**Rating:** 4
**Confidence:** 4

**Summary:**

This paper proposes a modality translation method from MRI to PET. The proposed method is based on the diffusion model and a Laplacian pyramid loss. The experiments are conducted on benchmark brain datasets and downstream tasks.

**Strengths:**

1. The proposed method is reasonable.

2. The method achieves promising performance in the experiments.

**Weaknesses:**

1. The major concern is that the technical contribution is limited. In general, the proposed method is a direct application of the diffusion model to the MRI-PET translation task.

2. The Laplacian pyramid approach has been a classic method in the generative model. Please refer to the following reference [a].

3. I wonder if some clinical information or domain knowledge can be introduced into the generative model.

[a] Deep generative image models using a Laplacian pyramid of adversarial networks. NeurIPS 2015.

**Questions:**

Please refer to the weaknesses.

---

### Official Review · Reviewer_4i67 · 2025-11-13

**Soundness:** 2
**Presentation:** 2
**Contribution:** 1
**Rating:** 2
**Confidence:** 4

**Summary:**

This paper proposes a 3D diffusion-based MRI-to-PET generation model designed to address the scarcity of paired MRI–PET data and the complexity of 3D PET synthesis. The method incorporates pre-training with style-transferred pseudo-PET images and a Laplacian pyramid loss for enhanced structural fidelity. The proposed approach was evaluated on the ADNI dataset for predicting AV45-PET scans from T1-weighted MRIs, where its performance was compared against GAN-based and vanilla diffusion baselines in terms of quantitative metrics (FID, SSIM, PSNR), qualitative realism, and downstream disease classification.

**Strengths:**

1. The idea of generating PET-like images from MRI-only data using MRI-to-PET style transfer based on a pre-trained VGG network and leveraging them for diffusion model pre-training is interesting.

2. The use of an additional loss (i.e, Laplacian pyramid loss) to capture multi-scale structural fidelity is a thoughtful design choice that enhances the model’s ability to preserve fine-grained anatomical details.

3. The study provides a comprehensive evaluation of MRI-to-PET synthesis, including quantitative metrics, qualitative visualization, and downstream clinical utility.

**Weaknesses:**

1. The major limitation lies in the insufficient review and comparison with prior studies in MRI-to-PET translation and medical image-to-image translation. The paper lacks discussion and experimental baselines against state-of-the-art (SOTA) medical image translation models, making it difficult to assess how much the proposed method advances beyond existing approaches.

2. The paper’s main novel components—style-transfer-based pre-training and the Laplacian pyramid loss—are not sufficiently motivated or contextualized. It is unclear what the term Laplacian pyramid loss precisely refers to, what its origin is, and which prior works inspired its adoption. The rationale for choosing this particular loss function should be clarified.

3. The evaluation is limited to AV45-PET from the ADNI dataset. While the method appears conceptually applicable to other PET modalities, this was not demonstrated. Hence, the generalizability of the approach remains uncertain based on the presented evidence.

4. The rationale underlying the use of style transfer—that PET and MRI share identical content but differ mainly in style—is not well justified. The paper would benefit from a clearer theoretical or empirical rationale for treating modality transfer as a style transfer problem.

**Questions:**

1. Could the authors provide more details about the diffusion model architecture and the conditioning mechanism used in MRI-to-PET generation? If a pre-implemented model or public library was utilized, please specify which one.

2. Have the authors tested the proposed model on other PET modalities such as amyloid or tau PET? How generalizable is the approach across different PET tracers?

3. What were the computational requirements for training? Specifically, how many epochs and how much training time were needed on an A100 GPU, and what were the batch size and memory usage? Since the model appears to perform diffusion modeling in the original 3D space, was GPU memory consumption manageable during training?

---

### Official Review · Reviewer_aMa7 · 2025-11-18

**Soundness:** 3
**Presentation:** 2
**Contribution:** 2
**Rating:** 4
**Confidence:** 4

**Summary:**

This paper proposes MRI2PET, a diffusion model framework for generating AV45-PET from T1 MRI, aimed at alleviating the problems of PET data scarcity, high cost, and difficulty in cross modal modeling of medical images. Experiments based on the ADNI dataset have shown that MRI2PET outperforms various GAN and diffusion baselines in indicators such as FID, SSIM, PSNR, and significantly improves performance in Alzheimer's classification and MMSE prediction, demonstrating the practical clinical value of synthetic PET in downstream tasks. The method consists of two core contributions:
1) The Style Transformer pre training strategy utilizes a large number of unpaired MRI images to stylize them into "PET like MRI", thereby constructing a pre training task that approximates the conversion of MRI to PET, allowing the model to learn cross modal structural mapping before limited real MRI-PET pairing training;
2) Laplacian Pyramid Loss， Add multi-scale detail constraints in addition to the standard noise prediction loss in the diffusion model to enhance the structural consistency and cross scale texture expression of generated PET.

**Strengths:**

1) The effectiveness of the diffusion model was improved by utilizing a pre training strategy with a large number of unpaired MRI scans.
2) The Laplacian pyramid loss enhances cross scale structural details and indeed brings quantitative benefits.

**Weaknesses:**

1) Insufficient innovation points: The style transfer technique and Laplace pyramid in the paper are existing technologies, but they are only application level innovations.
2) Abnormal experimental results: In Table 1, the results of other comparison methods are very poor, and many new technologies (such as diffusion model series) are actually much less effective than the old technology (GAN). There is a lack of relevant explanation here.
3) Effectiveness of ablation experiment: In Table 1, the results showed almost no decrease after removing the style transfer technique, and the loss was even better after removing the Laplacian pyramid, which is difficult to prove the effectiveness.
4) Insufficient downstream experiments: In the validation of downstream tasks, there is a lack of paired MRI and PET combined with a large number of MRI, and the results of MSE and Pearson correlation using only generated PET seem to be poor. Also, please unify the terminology for Synthetic PET and Generated PET.
5) Insufficient comparison method: The comparison method used is outdated and has not been compared with state-of-the-art diffusion models, and Vanilla Diffusion does not have any references.

**Questions:**

1) Given that the images generated by the diffusion model are real images with real noise added and prediction noise removed, can the Laplacian pyramid loss be replaced by the loss between images and noise?
2) LDM is more innovative than DDPM, why not use LDM as the basic model?
3) What is the motivation for style transfer?
4) Can cross dataset validation be performed?

---

### Note · Authors · 2025-12-05

I have read and agree with the venue's withdrawal policy on behalf of myself and my co-authors.